# Research on anchor chain visualization for a ship anchoring simulation training system

**Xiaobin Jiang**[ID][1]*, **Hongxiang Ren**[2], **Xin He**[3]

**1** Merchant Marine College, Shanghai Maritime University, Shanghai, China, **2** Marine Dynamic Simulation and Control Laboratory, Dalian Maritime University, Dalian, China, **3** Network Communication Research Center, Pengcheng Laboratory, Shenzhen, China

\* dmu_rhx@163.com

**Data Availability Statement:** All relevant data are within the paper and its Supporting Information files.

**Funding:** National High-tech Research and Development Program [No.2015AA016404],

## Abstract

In the development of ship anchorage training systems, the problems of low efficiency and poor fidelity exist in the simulation of flexible anchor chains, and a position-based dynamics (PBD) method is proposed to express the chain movement. To satisfy the requirements of simulating anchoring manipulation, the PBD method modifies the position of anchor chain particles by controlling constraints. Using the original distance constraint and bending constraint of the PBD approach, two novel constraints, namely, the long-range attachment (LRA) constraint and pin constraint, are developed to simulate the bending and stretching of the anchor chain. Simulation of ordinary ropes can be achieved using distance and bending constraints. The developed LRA constraint is capable of preventing anchor chain particles from being overstretched. Adoption of the pin constraint is proposed to integrate two particles into one to be calculated as an attempt to simulate the connection between the chain and the anchor. The continuous collision detection (CCD) constraint method considering friction and viscosity is used to detect collisions in the ship anchoring training system. Collision detection covers chain collisions with other objects and chains. Finally, the PBD method is more efficient and robust than the Newton method. Since it has sufficient visual plausibility and can realize real-time visualization, the simulation system developed by the PBD method effective for training crew members.

## Introduction

Anchoring refers to a common mode of ship handling. Since shipping companies do not have windlass devices for training, many crew members cannot skillfully manipulate the windlass. Accidents are often attributed to improper operation by crew(e.g., anchor loss, chain break and windlass burnout). In recent years, virtual reality technology has been advancing. The problem of the less training can be solved using virtual reality technology by developing a ship anchoring operation simulation system that exhibits immersion and interactivity.

The STCW (International Convention on Standards of Training, Certification and Watch keeping for Seafarers) Convention stipulates the standard requirements for correct anchoring procedures in Manila 2010 Amendment 2A-/1 [1]. In 2011, the DNV classification society

Fundamental Guidance Program for Provincial
Natural Science [No.20170540092]. Natural
Science Foundation of Liaoning Province.

**Competing interests:** The authors have declared
that no competing interests exist.

raised additional requirements for the "simulator to train anchoring operation" in a new standard of navigation simulators [2], including additional requirements of physical reality, behavioral realism and operating environment. The anchoring simulator is primarily used to train crews at the management level (e.g., captain or chief officer) and at the operation level (e.g., second officer or third crew) in anchoring, braking, and other anchoring operations [3].

The different phases of operating the anchor chain lead to the complexity in simulating the chain. Numerous scholars worldwide have studied the engineering aspects of ship anchoring manipulation. The scholars have primarily focused on whether the anchoring system is accurate and scientific [4], whereas they have rarely discussed the aspects of teaching and training. Few scholars have studied ship anchoring via three-dimensional visualization. To enhance reality during ship anchoring, we analyze the movement posture of the chain during anchoring, and then develop the ship handling simulation system. This paper purposes a PBD method to simulate the chain process, that is more comprehensive and reasonable than currently available method. Finally, three-dimensional visualization of ship anchoring is achieved.

## Related work

The most commonly used method to simulate ship anchoring system is based on force. The methods used to study anchoring systems can be split into static analysis methods and dynamic analysis methods. Static methods mainly includes the catenary method. The catenary method is extensively applied in the classification society [5]. The static analysis method is simple but not accurate. For dynamic analysis methods, there are numerous well-built methods for ship anchoring systems (e.g., the mass-spring method [6], the lumped mass method [7], the finite segment method [8], the finite element method [9], and the finite differences method [10]). In dynamic analysis methods, the external load of the anchor chain is transformed.

The major goal of chain simulation is to replicate anchor handling as accurately as possible. In contrast, the physical reality, behavioral reality and environmental reality are the critical factors affecting ship handling simulation systems. Indeed, the built model should be visually plausible.

Since Müller *et al.* [11] proposed PBD in 2007, the PBD method has been extensively used in interactive environments. The original formulation of PBD only considered soft bodies (e.g., cloths and inflatable balloons) [12]. Several works have recently been proposed to cover fluids [13], rigid bodies [14] and other objects. Rungjiratananon *et al.* [15] introduced a chain shape matching (CSM) model to simulate the tearing, twisting and flicking of strings. In the model, a string is represented as a chain of particles connected by segments. Kim *et al.* [16] proposed a long-range attachment (LRA) method capable of effectively and quickly achieving global scalability. An aerodynamic model is applied to cloth in [17]. There are extensive descriptions of this method and its derivatives, for example, in [18] and [19]. The PBD method is used to simulate ropes, cloth and deformable bodies, but it is rarely applied to the actual scene. In the ship anchoring operation, it is relatively difficult to simulate the retraction of the anchor chain. In this paper, we use the PBD method to simulate the process of chain release and retraction, realize three-dimensional visualization of mooring cable retraction and release, and finally, develop the ship anchoring operation simulation system combined with the established three-dimensional ship.

## Position-based dynamics

The chain is modeled as a set of constraints by PBD. Three steps in simulation iteration calculations are conducted in the simulation system:

- Moving the chain particles according to the velocity and external force;

- Moving the chain to satisfy the constraints;

- Performing time integration.

In the simulation system, the chain is represented by $N$ vertices and $M$ constraints. Each particle $i$ has a velocity $v_i$, position $x_i$ and mass $m_i$. A constraint $j \in [1, \ldots, M]$ includes

| cardinalities | $n_j$ |
| --- | --- |
| vector function | $C_j : \mathbb{R}^{3n_j} \to \mathbb{R}$ |
| set of particle indices | $\{i_1, \cdots, i_{nj}\}, i_k \in \{1, \cdots, N\}$ |
| stiffness parameter vector | $k \in \{0 \cdots 1\}$ |
| type | $equality$ or $inequality$ |

Constraint $j$ satisfies type $equality$ if

$$C_j(x_{i_1}, \cdots, x_{i_{n_j}}) = 0$$

If its type is $inequality$, it is defined by

$$C_j(x_{i_1}, \cdots, x_{i_{n_j}}) \geq 0.$$

Based on the data and time step $\triangle t$ given above, the simulation steps are expressed as algorithm 1.

In algorithm 1, the velocities and positions of the chain in (1)-(3) should be specified. In lines (5)-(6), a simple forward Euler integration is performed on the velocities and positions. The calculated new locations $p_i$ only act as prediction positions. Non-permanent external constraints (e.g., collision constraints) are generated at the beginning of each time step in line (7). The original and predicted positions are employed to perform continuous collision detection. The solver (8)-(10) then iteratively corrects the predicted positions, so they satisfy the $M_{coll}$ external as well as the $M$ internal constraints. Finally, the corrected positions $p_i$ are used to update the velocities and positions.

Algorithm 1 Position-based dynamics.
```
(1)  for all vertices i do
(2)     initialize x_i = x_i^0, v_i = v_i^0, w_i = 1/m_i
(3)  end for
(4)  end for
(5)     for all vertices i do v_i ← v_i + △tw_i f_ext(x_i)
(6)     for all vertices i do p_i ← x_i + △tv_i
(7)     for all vertices i do generate Collision Constraints (x_i → p_i)
(8)     loop solver Iterations times
(9)       project Constraints (C_1, ⋯, C_M + MColl, p_1, ⋯, p_N)
(10)    end loop
(11)    for all vertices i do
(12)      v_i ← (p_i − x_i)/△t
(13)      x_i ← p_i
(14)    end for
(15)    update velocity (v_1, ⋯, v_N)
(16) end loop
```

In algorithm 1, the symbol $p$ is adopted to distinguish the predicted positions from the positions of the previous time step. In the following, the symbol $x$ is employed for the positions of the particles. The problem that should be solved covers a set of $M$ equations for the $3N$

unknown position components. Vector $x$ is set as the concatenation $[x_1^T, \cdots, x_N^T]^T$, and $x$ and $M + M_{coll}$ constraints are set as the input conditions for all constraint functions $C_j$. The equation to be solved is expressed as

$$C_1(x) = 0, \cdots, C_M(x) = 0. \tag{1}$$

We use the nonlinear Gauss-Seidel iteration to solve the constraint equation and calculate the predicted position of the particles. The PBD method linearizes each constraint function separately. Each constraint equation can be solved separately, and each constraint produces a single scalar equation $C(x) = 0$ for all the particle locations associated with it. Given $x$, we attempt to find a correction $\triangle x$, so $C(x + \triangle x) = 0$. This equation can be approximated by

$$C(x + \triangle x) \approx C(x) + \bigtriangledown_x C(x) \cdot \triangle x = 0 \tag{2}$$

$\triangle x$ is restricted to be in the direction of $\bigtriangledown_x C$ to maintain linear and angular momentum directions to solve uncertainty problems. To achieve stability quickly, we have

$$\triangle x = \lambda \bigtriangledown_x C(x) \tag{3}$$

where $\lambda$ is the Lagrangian multiplier. Substituting Eq (3) into Eq (2), we can solve for $\lambda$

$$\triangle \lambda = -\frac{C(x)}{|\bigtriangledown_x C(x)|^2} \tag{4}$$

and substituting back into Eq (3), we obtain

$$\triangle x = -\frac{C(x)}{|\bigtriangledown_x C(x)|^2} \bigtriangledown_x C(x) \tag{5}$$

The above formula is the conventional iterative step for the solution of nonlinear equations given by a single constraint. Then, the final formula is yielded for $\triangle x$ of a single particle $i$

$$\triangle x_i = -s w_i \bigtriangledown_{x_i} C(x) \tag{6}$$

where

$$s = \frac{C(x)}{\sum_j w_j |\bigtriangledown_{xj} C(x)|^2} \tag{7}$$

where $s$ is the value of the scale factor, and $s$ is the same for all particles in the range of constraints. If the points have individual masses, the Newton-Raphson process is extended by weighting the corrections proportional to the inverse mass $w_i = 1/m$. After finding $\triangle x$, the current position is updated to $x \leftarrow x + \triangle x$. A new linear system is generated by evaluating $\bigtriangledown_x C_j(x)$ and $- C_j(x)$ in new locations. Subsequently, the process is repeated.

## Specific constraints

### Distance constraint

Distance constraints link pairs of particles together, forcing them to maintain a certain distance $d$ with each other. The distance constraint function is written as:

$$C(x_1, x_2) = |x_1 - x_2| - d \tag{8}$$

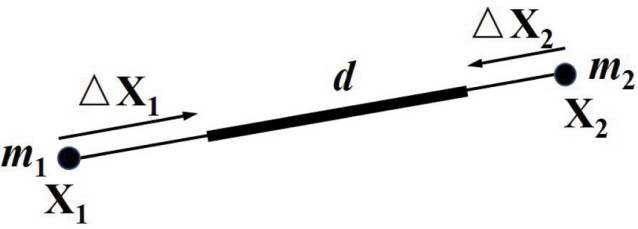

**Fig 1. Projection of the distance constraint.** $x_1$, $x_2$ are particle positions, $C(x_1, x_2)$ are constraint functions. The constraint function keeps the particles at a certain distance $d$ by directly moving the position of the particles.

The derivatives of the points are $\nabla_{x_1} C(x_1, x_2) = n$ and $\nabla_{x_2} C(x_1, x_2) = -n$ with $n = \frac{x_1 - x_2}{|x_1 - x_2|}$. According to inverse mass $w_i = 1/m_i$, the derivatives are weighted to obtain the final corrections

$$\triangle x_1 = -\frac{w_1}{w_1 + w_2}(|x_1 - x_2| - d)n \tag{9}$$

$$\triangle x_2 = +\frac{w_2}{w_1 + w_2}(|x_1 - x_2| - d)n \tag{10}$$

The distance constraint formulas for the projection proposed in [20]; and the particle moving direction are shown in Fig 1.

The distance constraint falls into the stretching constraint and compression constraint. For the stretching constraint, the chain tension is associated with the stiffness of the anchor chain. While the stiffness parameter $k_{stretch} \in (0, 1)$ has low values, the constraints will offer little resistance to exceed their rest length, thereby making the chain appear elastic. High values will make it harder for the constraints to stretch, as shown in Fig 2. The compression constraint is the slack of the anchor chain. The effect of different slack chains is shown in Fig 3. The lower the compression resistance value of the anchor chain is, the more incompressible it will be. The lighter the chain is, the more high slack values will benefit.

## Bending constraint

In addition to stretching resistance, bending should be simulated. Each bending constraint will work on three particles to arrange them in a straight line. For instance, the constraint will attempt to move $x_1$ along the red dashed line in Fig 4 to achieve chain resistant bending. The bending constraint function for each pair of adjacent straight lines $(x_1, x_2)$ and $(x_1, x_3)$ is expressed as [21]

$$C_{bend}(x_1, x_2, x_3) = arccos(n_1, n_2) - \varphi_0 \tag{11}$$

The actual angle $\varphi$ of the two lines is calculated by the angle between the normal of the two lines. The scalar $\varphi_0$ is the initial angle between two lines, and $k_{bend}$ refers to a global parameter defining the bending stiffness of the chain. The advantage of the bending constraint associated with the distance constraint between chain points $x_2$ and $x_3$ is that it is not determined by stretching. This is because the bending constraint is independent of hte chain lengths.

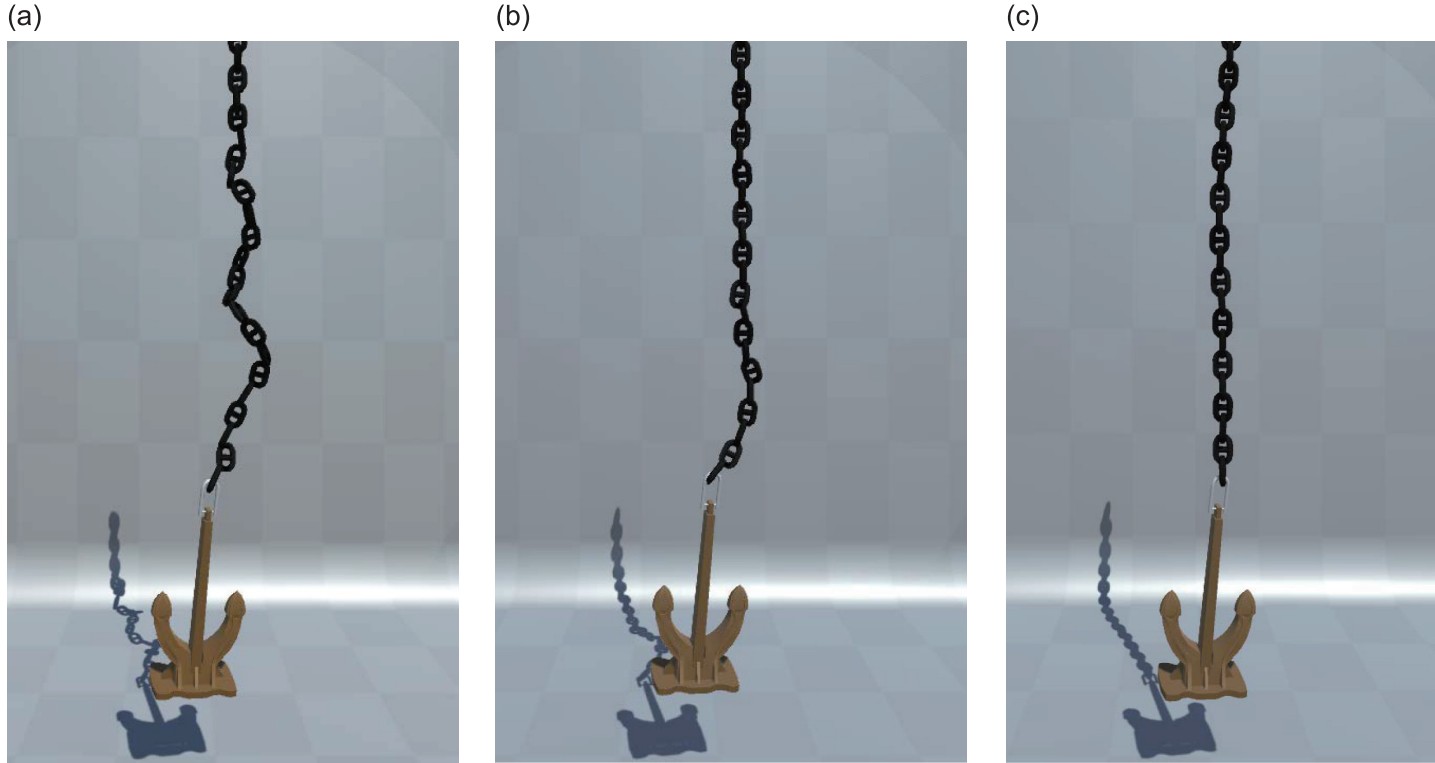

**Fig 2. Stretching constraint.** While $k_{stretch} = 0$, the chain will be more elastic. With the increase of $k_{stretch}$ it is harder to stretch.

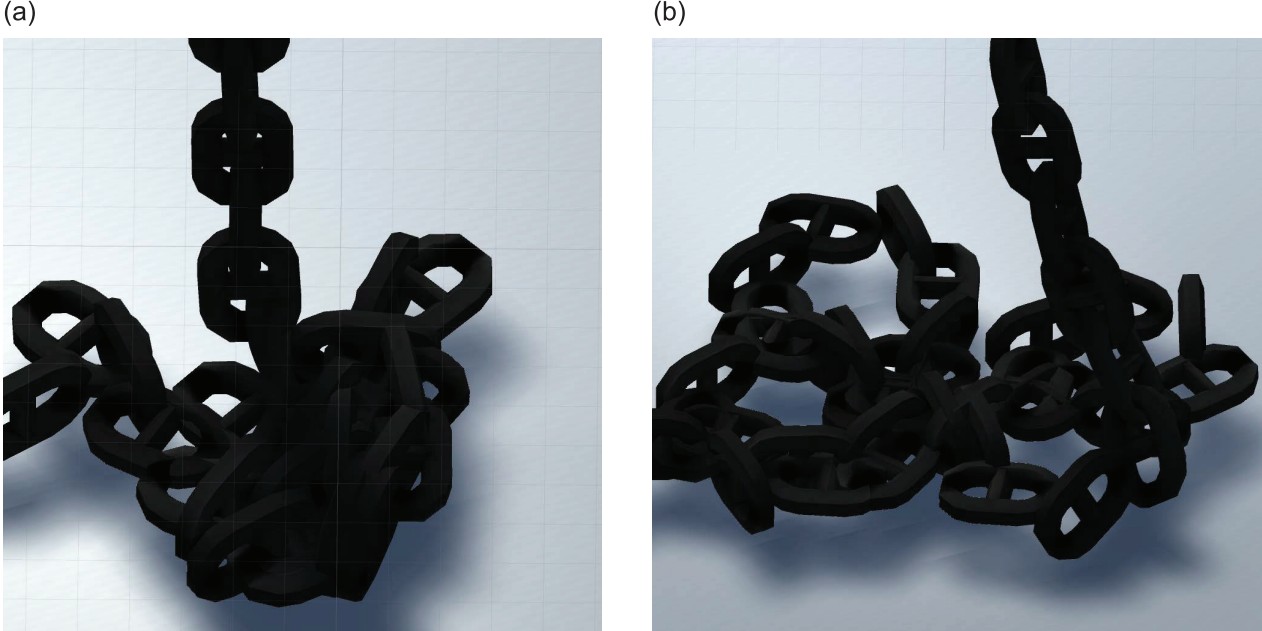

**Fig 3. Compression constraint.** While $k_{slack} = 1$, the chain will be more slacker. While $k_{slack} = 0$, the chain will be more incompressible.

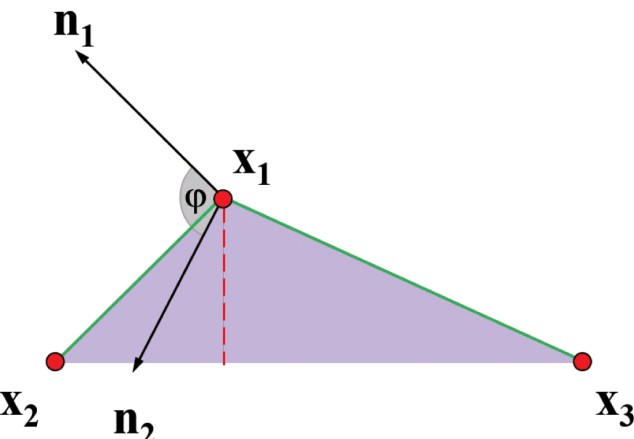

**Fig 4. Bending constraint.** $\varphi_0$ is the initial angle between the normal lines of a straight line $(x_1, x_2)$ and $(x_1, x_3)$.

According to $n_1 = \frac{x_1 \times x_2}{|x_1 \times x_2|}$, $n_2 = \frac{x_1 \times x_3}{|x_1 \times x_3|}$ and $\frac{d}{dx} arccos(x) = -\frac{1}{\sqrt{1-x^2}}$, we obtain the final correction $\triangle x_i$ of the particles

$$\triangle x_i = -\frac{3w_i}{\sum_j w_j} \frac{\sqrt{1-d^2}(arccos(d) - \varphi_0)}{\sum_j |q_j|^2} q_i \tag{12}$$

where, $q$ is the vertex position, and $d$ is the distance between particles.

The anchor chain has the characteristics of high tensile rigidity and high impact toughness. When the maximum bending amount is relatively low, the simulation effect is more in line with the actual situation of mooring line. We can specify the anchor chain with high bending resistance but low stretching stiffness to satisfy the actual situation.

### Long-range attachment constraint

On the whole, to converge the chain model, several iterations are required. The Gauss-Seidel iterative method addresses the constraints one by one, and the slow convergence results in the elastic distortion of the chain particles. Kim *et al.* [22] reported that the LRA method allows the cloth to converge with low iteration counts. The approach leverages the fact that stretching deformation always occurs when the cloth is attached. LRA constraints are highly similar to distance constraints, which are all used to reduce the stretching of slender bodies. However, these constraints are designed to reduce the overstretching of long-distance constraints. The LRA method is employed to prevent the chain from stretching at low iterations.

If one end of the chain is attached at the hawsehole, the distance between the hawsehole and the chain particle for all chain particles cannot be larger than the initial length. Therefore, the LRA method is formalized as follows. For each unconstrained particle $i$, the initial distance $r_i$ is precalculated from the chain particle to the hawsehole. During the simulation, if the particle is within this distance, it is allowed to move freely. If the particle moves beyond this limit, it is projected onto the surface of a sphere with radius $r_i$. By enforcing this constraint, tensile pressure waves are allowed to propagate from the hawsehole to all free chain particles in a single step.

The LRA method is shown in Fig 5 to simulate an inextensible chain attached to a hawsehole (red point). Each chain particle is constrained into the hawsehole centered on the attachment point. The radius of the sphere is the initial distance from the particle to the attachment

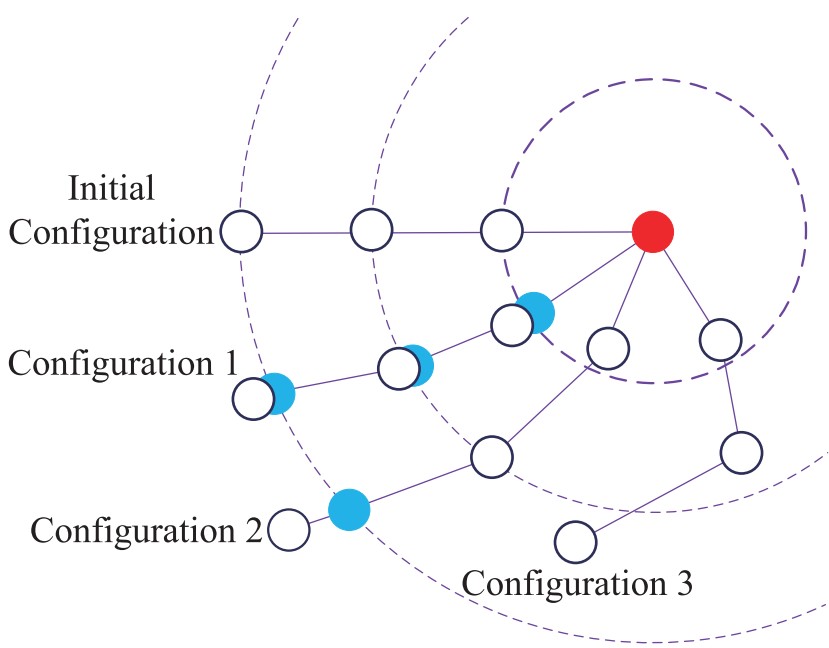

**Fig 5. LRA constraint.**

point. For each configuration, target chain positions are marked in blue when particles should be projected. The chain particles in the constraint spheres are allowed to move freely, and particles outside the constraint sphere need to be projected onto the sphere. For example, in configuration 1, the chain particles will move to the blue point, and in configuration 3, the chain particles will move freely. Fig 6a–6f shows the status that the chain attached to the anchor is

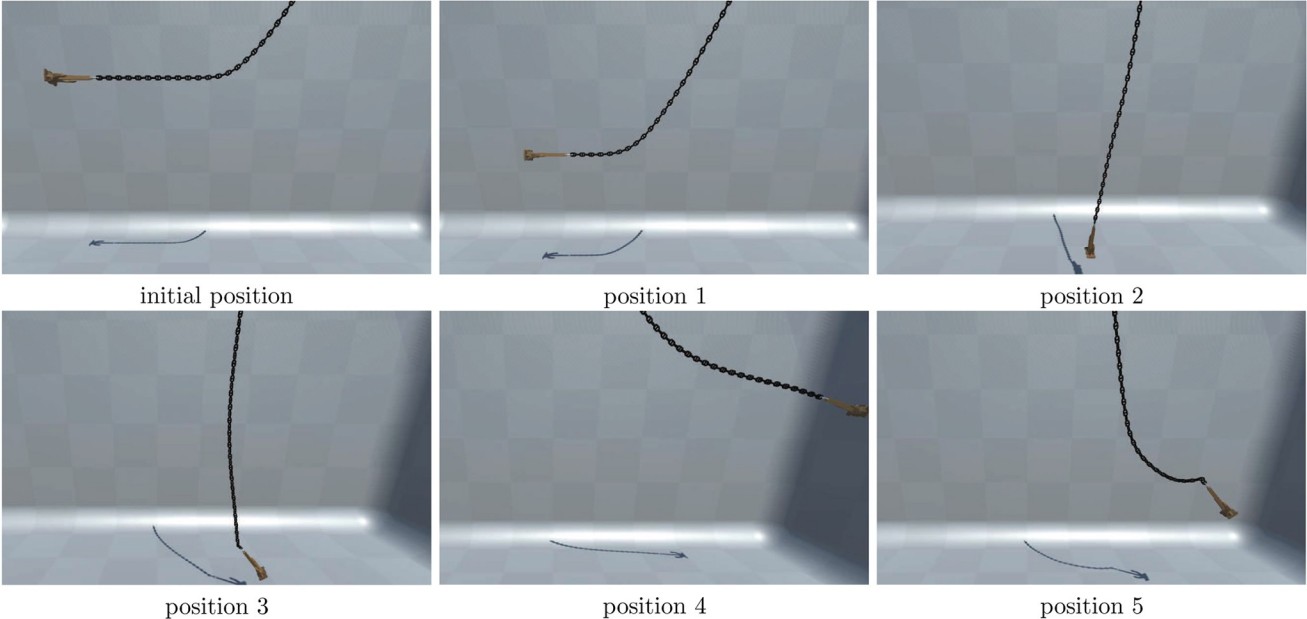

initial position  position 1  position 2

position 3  position 4  position 5

**Fig 6. LRA to simulate an inextensible chain with the anchor.**

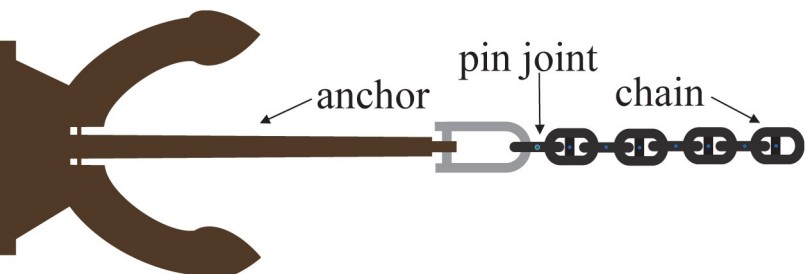

**Fig 7. Pin constraint.**

simulated in the free-falling process of the anchor chain from horizontal. The motion process is similar to that of a pendulum. The position of the anchor chain is intercepted regularly, and the anchor chain finally stops at the vertical ground state.

## Pin constraint

The anchor and chain are connected. Pin constraints help attach chain links to an anchor, i.e., two objects share one particle, and they are connected by the pin connection. A two-way energy exchange channel is established: chain link movements will affect the anchor, and anchor movements will affect the chain link. The hinged restraint exerts a damping force on the particles and hinge objects to maintain their relative positions [23]. Fig 7 shows the anchor and chain connected together by the pin constraint.

## Collision constraint

The position based approach can achieve a collision response by the continuous collision detection (CCD) method. In line (7) of Algorithm 1, the collision constraints $M_{coll}$ are generated. The number of colliding vertices determines the collision constraints $M_{coll}$. The velocity of each vertex of the generated collision constraint is perpendicular to the collision normal, as reflected in the normal direction of the collision. Using our simulator to simulate two objects, the response to dynamic colliding can be correct. Chain particles are capable of colliding and reacting with other objects in the scene; they can collide with each other as well.

**Particle collision.** Collisions between particles can be processed by linearizing and introducing a contact plane, similar to how the environment is processed. However, the nonlinear nature of the constraint is usually more robustly maintained as

$$C(x_i, x_j) = |x_{ij}| - (r_i - r_j) \geq 0 \tag{13}$$

where $r_i$ and $r_j$ denote the radii of the chain particles. The collision constraint is capable of simulating granular-like materials as proposed in [22]. Fig 8 shows particle collisions with and without self-collision, and there are 100 particles on both sides. In the figure, we can see that when considering collisions between particles, the particles will pile up. Without particle self-collision, all the particles lie on the ground.

The number of particles used to generate the chain is determined. A value of 1 will create particles with a spacing equal to the radius of particles, allowing them to overlap and create a tight representation of the chain. A value of 0.5 will maintain the distance of particles, so they just contact each other. Lower values leave lagger gaps between particles (Fig 9).

**Environment collision.** Collisions between particles and kinematic objects (e.g., deck, seabed or dock) can be addressed first by detecting the contact planes for each particle.

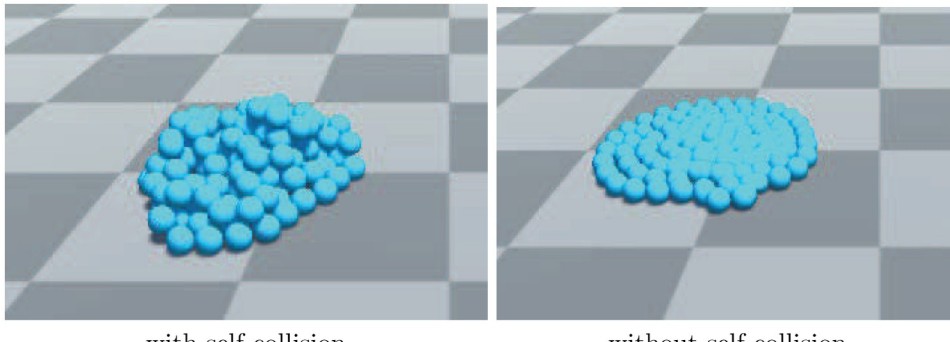

**Fig 8. Particles with and without self-collision.**

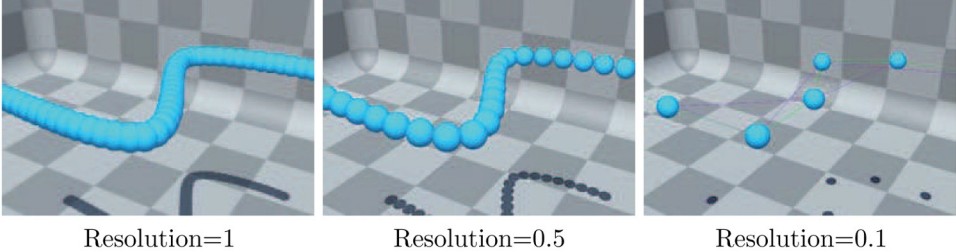

**Fig 9. Particle collisions with each other with different resolution.**

Subsequently, for each contact plane normal $n$, nonpenetration constraint is introduced into the system

$$C(x) = n^T x - d_{rest} = 0 \tag{14}$$

where, $d_{rest}$ denotes the distance that the chain particles should be kept on the kinematic objects.

The radius of the chain particles should be set to enhance the authenticity of the system. Once the objects contact each other, a collision response is generated. According to the material properties of the particles and the collision object, different collision responses are generated. The particle material properties include friction, viscosity, etc. When pulling the chain, the friction and viscosity between the chain particles and kinematic objects are considered. In Fig 10, the red arrow represents the velocity of the particle upon collision. This velocity consists of two components: one along the contact normal, and the other along the contact tangent. The blue arrow refers to the force acting on the particle to cancel its normal velocity. The purple arrow indicates the friction force applied to the particles, which eliminates a certain proportion of the tangential velocity. The green arrow indicates the stickiness force.

The constraint function ensures that $q$ remains above the chain of triangle $x_1$, $x_2$, and $x_3$ with thickness $h$.

$$C(q, x_1, x_2, x_3) = (q - x_1)n - h \tag{15}$$

Collision helps set the collision material and thickness. Positive thickness values leave a gap between the collider surface and colliding particles, while negative values cause the particles to

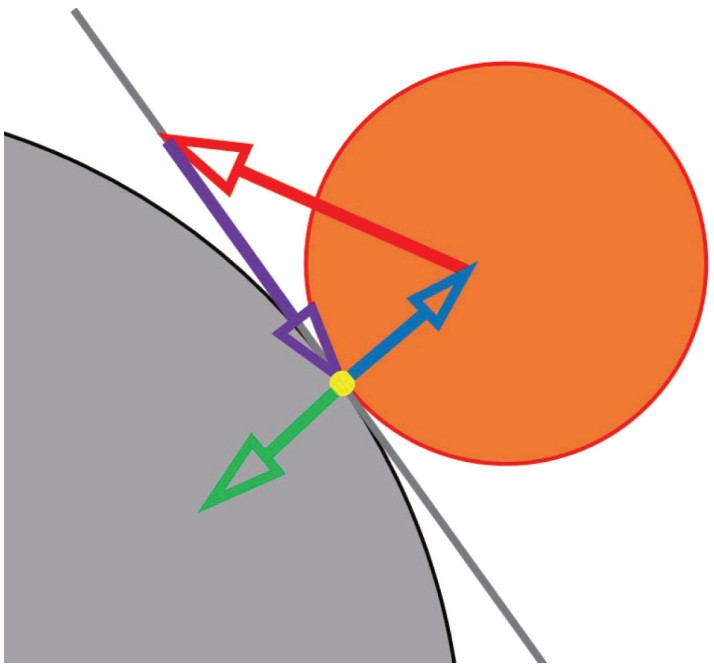

**Fig 10. Particle collision with a rigid body.**

"sink" into the collider before actually hitting its surface, as shown in Fig 11. The effect of particles with different thickness values colliding with the cube is shown in Fig 12, which is 0.08, 0 and -0.05 from left to right.

## Results

The ship handling simulation system is mainly used to model and simulate anchoring operation in 3D, and to study the dynamic response of ship anchoring. The 3D model of the whole ship in the simulation system consists of 20 million surfaces. When the anchor chain

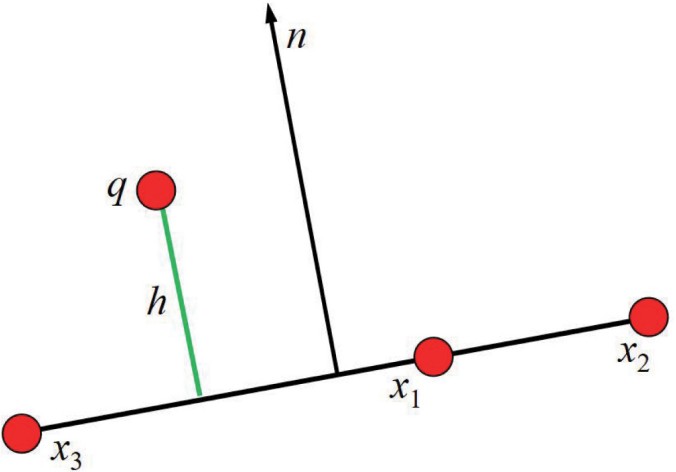

**Fig 11. Constraint with chain particle thickness.**

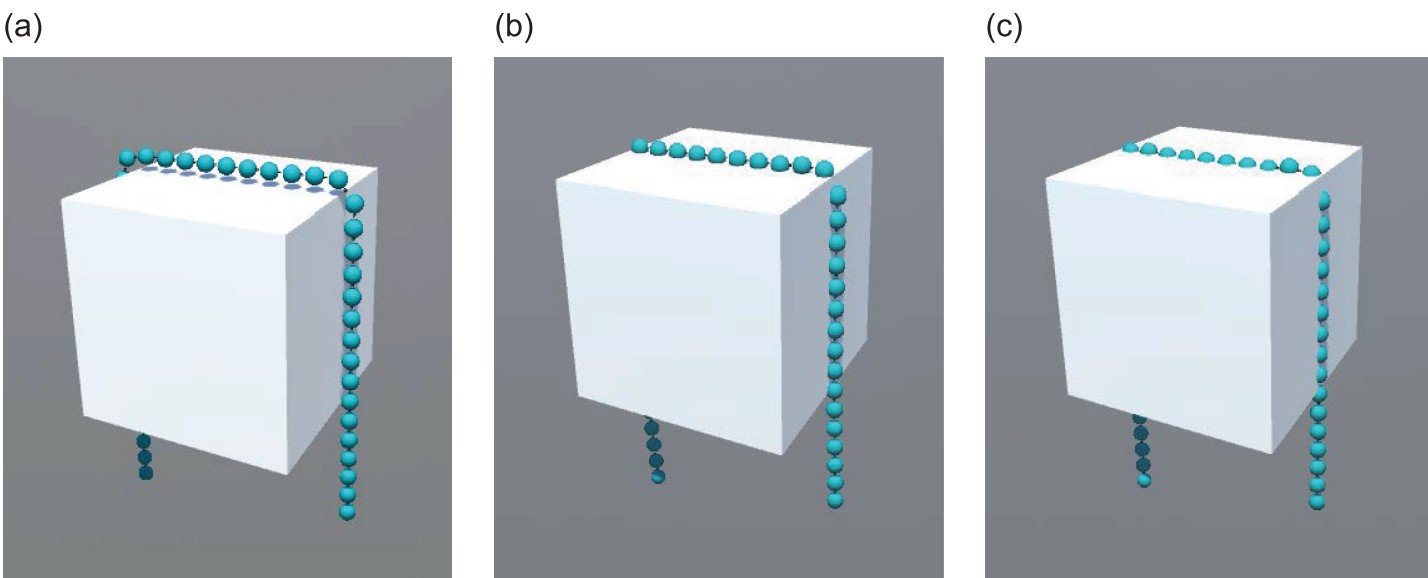

**Fig 12. Collision effect of anchor chain particles and a cube.**

simulation module is not added, the rendering speed will be over 100 fps. The stiffness of the model is dependent not only on the user-defined stiffness parameter but also on the time step size and the number of solver iterations. The chain is split into 1000 particles. Constraint coefficients and iteration number employed in the simulation system include: $k_{stretch} = 0.99$, $k_{bending} = 0.02$, $k_{LRA} = 1$, $k_{pin} = 1$, $n_{stretch} = 10$, $n_{bending} = 8$, $n_{LRA} = 3$, $n_{pin} = 5$. When subjoining the anchoring manipulation module under the above conditions, the rendering speed reaches over 80 fps, satisfying the real-time requirements. All the test scenarios presented in this section run on Windows 10. The program development software employs Unity3d and Visual Studio 2015, and an Intel(R) Core i7 CPU, and a GT 730 graphics card is make up the hardware environment of the system.

Fig 13 presents the visualization effect of the ship anchoring operation. In the figure, the anchor chain is fully recovered to the chain locker. Fig 13(a) shows the initial state of starting the chain release; Fig 13(b) shows part of the chain that is released; and Fig 13(c) shows the anchor thai is thrown out. Fig 13(d), 13(e) anf 13(f) shows the corresponding effects of the anchor chain from the outside perspective of the ship. Fig 13(g), 13(h) and 13(i) shows the final effect of ship mooring maneuvering based on the ship motion.

Fig 14(a) suggests comparing the relationship between the relative error and the iteration number using the Newton method. The relative error reported in Fig 14 is defined as:

$$\delta = \frac{g(x_i) - g(x*)}{g(x_0) - g(x*)} \tag{16}$$

where $x_0$ denotes the initial guess, $x_i$ is the current iteration, and $x^*$ is the final solution. With the same number of iterations, the Newton method converges faster than our Gauss-Seidel iterative method, which, however, does not reflect the cost of each iteration. This is because each Newton iteration requires a solution to the varying linear system. The relationship between the calculation time and relative error is considered, as shown in Fig 14(b). The figure suggests that the relative error obtained using the PBD method is smaller than that obtained using the Newton method simultaneously.

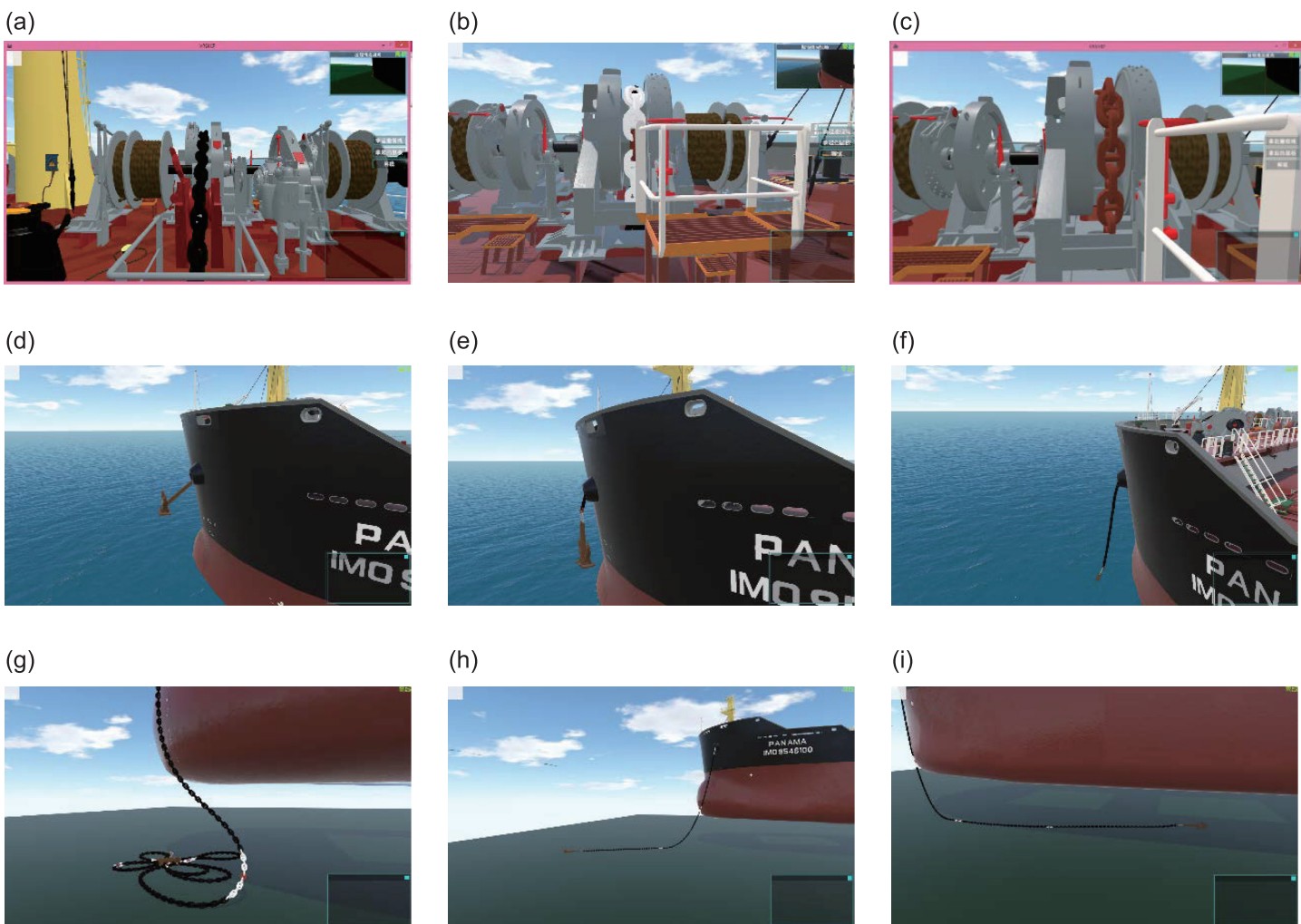

**Fig 13. Anchoring operation visualization effect.**

Using the PBD method in Fig 14, the calculation results are achieved, as listed in Table 1. The table suggests that 10 iterations will cost 18 ms, and that the relative error is $1.66 * 10^{-4}$, satisfying the requirements of simulation accuracy. However, 10 iterations completed using the Newton method, cost 1816 ms. The PBD method is capable of better satisfying the requirements of the simulation training system at a lower number of iterations and exhibits a more stable performance than Newton method. From the perspective of real-time simulation, the PBD method shows more advantages in interactive applications.

## Conclusions

Given the development of the ship handling simulation system, the PBD method is adopted to simulate the anchoring maneuver. During development, several specific constraints are employed based on the PBD framework to project the chain particles to the effective position. In the calculation of collision detection in the scene, the CCD method considering friction and viscosity is employed to achieve collision detection in the ship anchoring simulation system. The penetration problem of the anchor chain with the ship deck, dock and seabed is solved. Three-dimensional visualization simulation of the ship anchoring operation is achieved. The

(a)

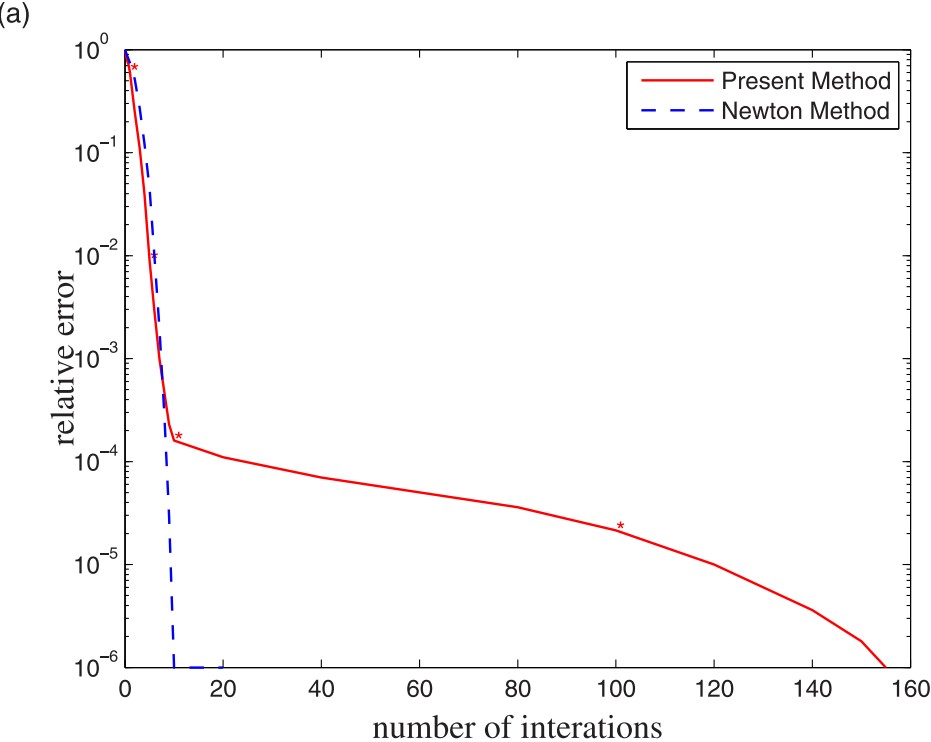

(b)

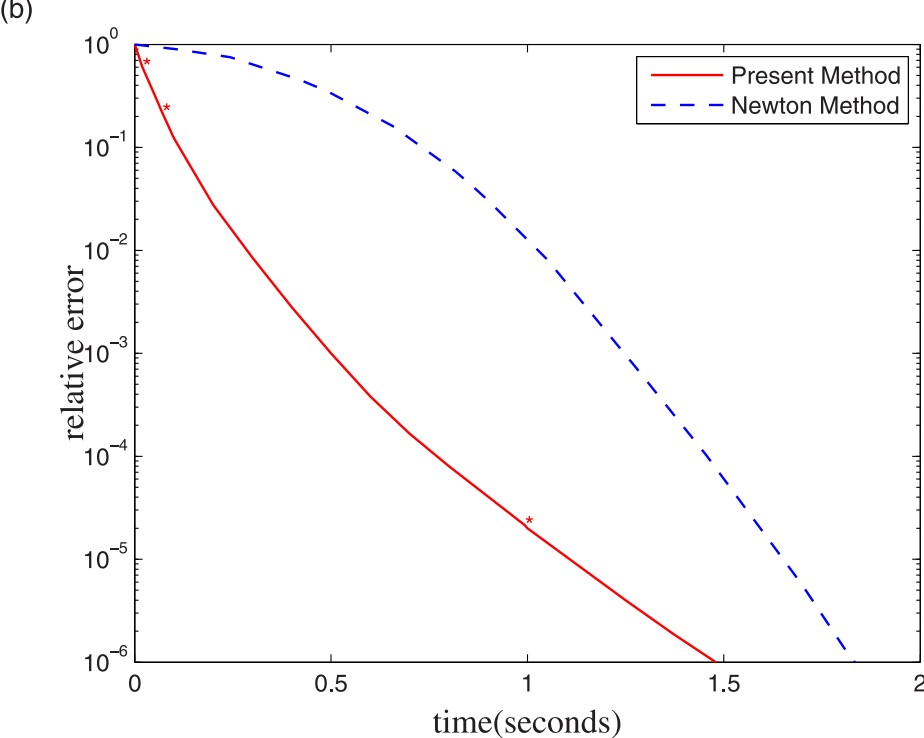

**Fig 14. Comparison of relative error vs. iteration count and time.**

**Table 1. Performance results based on the PBD method.**

| Iterations | Time(ms) | Relative Error |
|:---:|:---:|:---:|
| 1 | 18 | $6.11 \cdot 10^{-1}$ |
| 5 | 28 | $9.02 \cdot 10^{-3}$ |
| 10 | 70 | $1.66 \cdot 10^{-4}$ |
| 100 | 972 | $2.15 \cdot 10^{-5}$ |

optimized PBD method is more efficient than Newton's method, exhibiting sufficient visual plausibility. The operation of the ship anchoring simulation system is consistent with the actual situation, and it is repeatable, significantly saving costs and lowering risks. The PBD method acts as a novel way to achieve the development of three-dimensional ship anchoring simulation system. The built anchor chain simulation model is important for guiding ship anchoring manipulation simulation. However, there are also some disadvantages. The elastic stiffness of the chain is very large, although the stiffness of the model can be reduced by the LRA method, it cannot be completely removed. Decoupling these parameters as well as adaptive time stepping are problems and important topics for future work.

## Supporting information

**S1 Fig.**
(JPG)

**S1 Video.**
(WMV)

**S1 Graphical abstract.**
(PDF)

## Author Contributions

**Conceptualization:** Hongxiang Ren.

**Data curation:** Xiaobin Jiang, Xin He.

**Funding acquisition:** Hongxiang Ren.

**Investigation:** Xiaobin Jiang.

**Methodology:** Xin He.

**Supervision:** Hongxiang Ren.

**Validation:** Xin He.

**Visualization:** Xiaobin Jiang.

**Writing – original draft:** Xiaobin Jiang.

**Writing – review & editing:** Hongxiang Ren.

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
