## [Decision Letter · Decision Letter 0]

22 Jan 2020

PONE-D-19-25283

Research on anchor chain visualization of ship anchoring simulation training system

PLOS ONE

Dear Dr. Jiang,

Thank you for submitting your manuscript to PLOS ONE. After careful consideration, we feel that it has merit but does not fully meet PLOS ONE’s publication criteria as it currently stands. Therefore, we invite you to submit a revised version of the manuscript that addresses the points raised during the review process.

We would appreciate receiving your revised manuscript by Mar 07 2020 11:59PM. To enhance the reproducibility of your results, we recommend that if applicable you deposit your laboratory protocols in protocols.io, where a protocol can be assigned its own identifier (DOI) such that it can be cited independently in the future. For instructions see: http://journals.plos.org/plosone/s/submission-guidelines#loc-laboratory-protocols

We look forward to receiving your revised manuscript.

Kind regards,

Myung-Il Roh, Ph.D.

Academic Editor

PLOS ONE

Journal Requirements:

2. We note that your manuscript does not contain a labelled Methods or Results section. Please ensure that you label these sections appropriately, and include the relevant information in each. Please include in the Methods section information on the simulations run, and any data or parameters used.

4. We note that the Figures in your submission contain copyrighted images. All PLOS content is published under the Creative Commons Attribution License (CC BY 4.0), which means that the manuscript, images, and Supporting Information files will be freely available online, and any third party is permitted to access, download, copy, distribute, and use these materials in any way, even commercially, with proper attribution. For more information, see our copyright guidelines: http://journals.plos.org/plosone/s/licenses-and-copyright.

1.         You may seek permission from the original copyright holder of Figures 1-14 to publish the content specifically under the CC BY 4.0 license. 

Additional Editor Comments:

Please prepare the revision of your paper. All comments from the reviewers should be fully addressed in the revision.

Reviewers' comments:

Reviewer's Responses to Questions

**Comments to the Author**

1. Is the manuscript technically sound, and do the data support the conclusions?

Reviewer #1: Partly

Reviewer #2: Yes

2. Has the statistical analysis been performed appropriately and rigorously? 

Reviewer #1: I Don't Know

Reviewer #2: Yes

3. Have the authors made all data underlying the findings in their manuscript fully available?

Reviewer #1: Yes

Reviewer #2: Yes

4. Is the manuscript presented in an intelligible fashion and written in standard English?

Reviewer #1: No

Reviewer #2: No

5. Review Comments to the Author

Reviewer #1: The author tackled anchor handling simulation which is one of the most complicated problem in shipbuilding. Many well-known methods such as PDB method and LRA method are all included. However, the discussion is weak and the grammar should be more thoroughly checked.

Reviewer #2: This study presents the visualization method of ship anchoring simulation. The method is based on the Position-Based Dynamics, and shows how to define the constraints of the ship anchoring simulation. The study needs to cover the following comments for the acceptance in PLOS ONE journal.

1. The manuscript needs editing to English grammar and sentence for clear understanding to the reader.

2. What is the ship handling simulation system? It is mentioned that the system has 20 million surfaces. The study present position-based dynamics, that means it is based on the mesh. It needs more explanations.

3. p. 7. Line 129. What is the actual situation and how the chain can be specified like that?

4. p. 10. Line 194. It is not easy to understand why the friction force should be removed.

5. p.11. Line 225. How to define the x0? Is it related to time step?

6. p.12. Line 246. It is mentioned that the seabed penetration problem is solved but the problem is not mentioned in the manuscript. If the problem can be solved with the environment collision, it should be discussed in the section.

7. The simulation speed and error between PBD method and Newton method in the discussion. However, the verification of the simulation result should be added in the discussion. It is necessary to show the comparison between the exact solution and the simulation result.

6. PLOS authors have the option to publish the peer review history of their article (what does this mean?). If published, this will include your full peer review and any attached files.

Reviewer #1: No

Reviewer #2: No

---

## [Author Response · Author response to Decision Letter 0]

3 Jun 2020

We use the program development software employee Unity3D and Visual Studio 2015 to create the Figs 2, 3, 6, 7, 8, 9, 12, 13, the supporting information file "chain.wmv," and the figure file "Graphical Abstract.pdf."

We use professional scientific editing service by American Journal Experts (AJE) of my manuscript. We visit the AJE website (http://learn.aje.com/plos/), then we carefully check and update the grammar one more time before submitting.

---

## [Editor Report · Decision Letter 1]

12 Jun 2020

PONE-D-19-25283R1

Research on anchor chain visualization for a ship anchoring simulation training system

PLOS ONE

Dear Dr. Jiang,

Thank you for submitting your manuscript to PLOS ONE. After careful consideration, we feel that it has merit but does not fully meet PLOS ONE’s publication criteria as it currently stands. Therefore, we invite you to submit a revised version of the manuscript that addresses the points raised during the review process.

We look forward to receiving your revised manuscript.

Kind regards,

Myung-Il Roh, Ph.D.

Academic Editor

PLOS ONE

Additional Editor Comments (if provided):

The authors did not address all the comments from the reviewers. They responded to only two comments.

Please respond to all comments of two reviewers. Without this, I can not proceed to the next step.

You will have to prepare Author's response to the reviewers' comments.

---

## [Author Response · Author response to Decision Letter 1]

25 Jun 2020

Answer: The names of colleagues editing the manuscript include Hongxiang Ren, Xin He, etc. The professional services of editing manuscripts are “Zibo Yimore Translation CO.LTD” editing company.

Answer: We uploaded a high lighting manuscript as a *supporting information* file. 

Answer: We uploaded a clean copy of the edited manuscript as the new *manuscript* file.

---

## [Decision Letter · Decision Letter 2]

30 Jul 2020

Research on anchor chain visualization for a ship anchoring simulation training system

PONE-D-19-25283R2

Dear Dr. Jiang,

We’re pleased to inform you that your manuscript has been judged scientifically suitable for publication and will be formally accepted for publication once it meets all outstanding technical requirements.

Kind regards,

Myung-Il Roh, Ph.D.

Academic Editor

PLOS ONE

Additional Editor Comments (optional):

No more comment

Reviewers' comments:

Reviewer's Responses to Questions

**Comments to the Author**

1. If the authors have adequately addressed your comments raised in a previous round of review and you feel that this manuscript is now acceptable for publication, you may indicate that here to bypass the “Comments to the Author” section, enter your conflict of interest statement in the “Confidential to Editor” section, and submit your "Accept" recommendation.

Reviewer #1: All comments have been addressed

Reviewer #2: All comments have been addressed

2. Is the manuscript technically sound, and do the data support the conclusions?

Reviewer #1: Yes

Reviewer #2: Yes

3. Has the statistical analysis been performed appropriately and rigorously? 

Reviewer #1: Yes

Reviewer #2: Yes

4. Have the authors made all data underlying the findings in their manuscript fully available?

Reviewer #1: Yes

Reviewer #2: Yes

5. Is the manuscript presented in an intelligible fashion and written in standard English?

Reviewer #1: Yes

Reviewer #2: Yes

6. Review Comments to the Author

Reviewer #1: The author answered all questions appropriately. Everything I was curious about was resolved. Thank you for authors' effort.

Reviewer #2: All my comments are covered by the authors.

This manuscript can be accepted to the PLOS ONE Journal.

7. PLOS authors have the option to publish the peer review history of their article (what does this mean?). If published, this will include your full peer review and any attached files.

Reviewer #1: No

Reviewer #2: No

---

## [Editor Report · Acceptance letter]

24 Sep 2020

PONE-D-19-25283R2 

Research on anchor chain visualization for a ship anchoring simulation training system 

Dear Dr. Jiang:

I'm pleased to inform you that your manuscript has been deemed suitable for publication in PLOS ONE. Congratulations! Your manuscript is now with our production department. 

Kind regards, 

on behalf of

Prof. Myung-Il Roh 

Academic Editor

PLOS ONE